# The zona incerta negatively regulates the red nucleus during movement cued by sound signals

Liang Chen[1,2]*, Xinxing Wang[1], Hanxiao Liu[1], Chenzhao He[2], Allen P. F. Chen[1,3], Lu Chen[1], Thomas A. Kim[1,3], Qiaojie Xiong[1]*

1 Department of Neurobiology and Behavior, SUNY Stony Brook, Stony Brook, New York, United States of America, 2 Department of Human Anatomy and Histoembryology, School of Basic Medical Sciences, Fudan University, Shanghai, China, 3 Medical Scientist Training Program, Renaissance School of Medicine at SUNY Stony Brook, Stony Brook, New York, United States of America

* qiaojie.xiong@stonybrook.edu (QX); liangchen@fudan.edu.cn (LC)

## Abstract

Auditory signal-cued behaviors rely on a sophisticated neural network. While extensive research has focused on auditory processing and decision-making, the neural circuits governing motor coordination for goal-directed actions remain poorly understood. The red nucleus (RN) is essential for motor coordination, whereas the zona incerta (ZI) plays a key role in modulating sensorimotor circuits. Using tetrode recordings and optogenetics, we investigated the ZI-RN circuit's role in an auditory-cued decision task. RN neurons were preferentially activated when mice moved to the contralateral port for a reward, and optogenetic activation biased choices toward the contralateral side. Notably, parvalbumin-positive ZI neurons projected to the RN and negatively regulated movement coordination. These findings reveal an inhibitory ZI-RN circuit that shapes auditory-cued, goal-directed movement.

## Introduction

Sensory sensations provide necessary inputs to the central nervous system from our surroundings and internal organs to guide our body to take accurate actions [1]. Lines of recent studies from our group and others have found that the auditory signal processing pathways play a pivotal role in associating the auditory signals with movements for rewards in an auditory frequency discrimination task [2–6]. However, neural circuits underlying the motor coordination in this sensory-cued goal-directed behavior remain elusive.

The red nucleus (RN) in the ventral midbrain receives projection from motor cortex, basal ganglia, thalamus, and cerebellum, and connect to the spinal cord for controlling motor and non-motor behaviors [7]. It has been redeemed as a key node in goal-directed movements, particularly in limb control and fine motor coordination [8,9]. This important motor function of the RN motivated us to explore whether the RN possibly regulates movements toward the rewards guided by the auditory decisions.

In parallel, emerging evidence suggests that the zona incerta (ZI) plays a critical role in modulating sensorimotor circuits [10–19]. The ZI is known to exert widespread inhibitory

**Data availability statement:** All relevant data are within the paper and its Supporting Information files.

**Funding:** This work was supported by the SUNY Stony Brook internal fund to Q.X., and the Thomas Hartman Center for Parkinson's Research to Q.X. The funders had no role in study design, data collection and analysis, decision to publish, or preparation of the manuscript.

**Competing interests:** The authors have declared that no competing interests exist.

**Abbreviations:** AAV, adeno-associated virus; ChR2, channel rhodopsin 2; GFP, green fluorescence protein; IPSC, inhibitory postsynaptic current; Lhx6, LIM homeobox protein 6; PV, parvalbumin; RN, red nucleus; ROC, receiver operating characteristic; SNr, Substantia nigra pars reticulata; SST, somatostatin; STN, Subthalamic nucleus; WT, wild type; ZI, zona incerta.

control over various brain regions, including those involved in motor planning and execution. Notably, the ZI has been shown to project to the RN, raising the possibility that it may regulate RN activity during goal-directed behaviors.

To determine the role of ZI-RN circuit in sensory-cued goal-directed movement, we examine RN neuronal activities from mice performing an auditory-cued two-alternative force choice task. We found that a large group of RN neurons increased their firing rates during the goal-directed movement with contralateral preference. Optogenetic manipulations of these neurons biased mice performance in the task. We further found that the ZI has inhibitory projection to the RN, and optogenetic manipulation of three types of ZI inhibitory neurons have different effects on the auditory-guided movements in the task. Altogether, our study demonstrates an inhibitory circuit from the ZI to the RN that modulates the auditory-cued goal-directed movement.

## Results

### Activation of excitatory neurons in the red nucleus promoted the contralateral movement choice in the auditory task

To determine the role of the red nucleus (RN) in the auditory-cued goal-directed behavior, we recorded RN neuronal activities in mice performing an auditory frequency discrimination task as illustrated in Fig 1A or previously described [3]. In brief, mice learned to self-trigger an auditory cue by poking their noses into the center port, and then moved to one of the side ports for water reward depending on the tone frequencies in the cues (low tones to left port and high tones to right port). Mice's performance was driven by the water reward because they were water restricted, and they stopped task performance quickly if the water reward was omitted. We performed tetrode recordings in the left RN of adult wild type mice (WT, C57BL/6) and isolated individual excitatory units based on the waveforms of the action potentials as shown in Fig 1B. We aligned the activities of excitatory units to the initiation of movements when mice withdrew from the center port (Time from CenterOut) and moved to the side ports for water reward. The activity of a single neuron during trials has been exampled in Fig 1C. Among the 198 units of excitatory neurons recorded from the left RN of 5 mice (42 from mouse 1, 35 from mouse 2, 58 from mouse 3, 28 from mouse 4, 35 from mouse 5), we found that 61 units were preferentially activated during the movements from the center port toward the right-side port, referring to the contralateral movement (Fig 1D and 1E). These neurons showed the same activity patterns in both correct and error trials (S7A Fig) when they moved to the same side ports, indicating that the RN neuronal activity is correlated with the side of the movement (movement decision) rather than auditory cue identity (sensory representation). In contrast, there were only 13 units showing an ipsilateral preference. Importantly, similar analyses of inhibitory neurons revealed that there was no obvious side preference (S1A and S1B Fig). To analyze whether the 198 units exhibit any preferences in tone frequencies, we aligned the unit activity to the onset of auditory cues (Time from SoundOn). As shown in S2 Fig, only 19 of the 198 units exhibited potential preference to either the high ($n = 15$) or low ($n = 4$) frequency tones, suggesting that RN excitatory neurons did not exhibit a preferential activation to the frequencies of auditory cues. Together, these results suggest that excitatory neurons in the RN may be preferentially involved in contralateral movements in the task.

We next examined whether excessive activation of RN excitatory neurons right after the CenterOut impacts the movements towards the reward ports. To specifically and temporally activate RN excitatory neurons, we employed the optogenetic method as previously described [6]. In a cohort of adult WT mice, we injected adeno-associated virus (AAV)

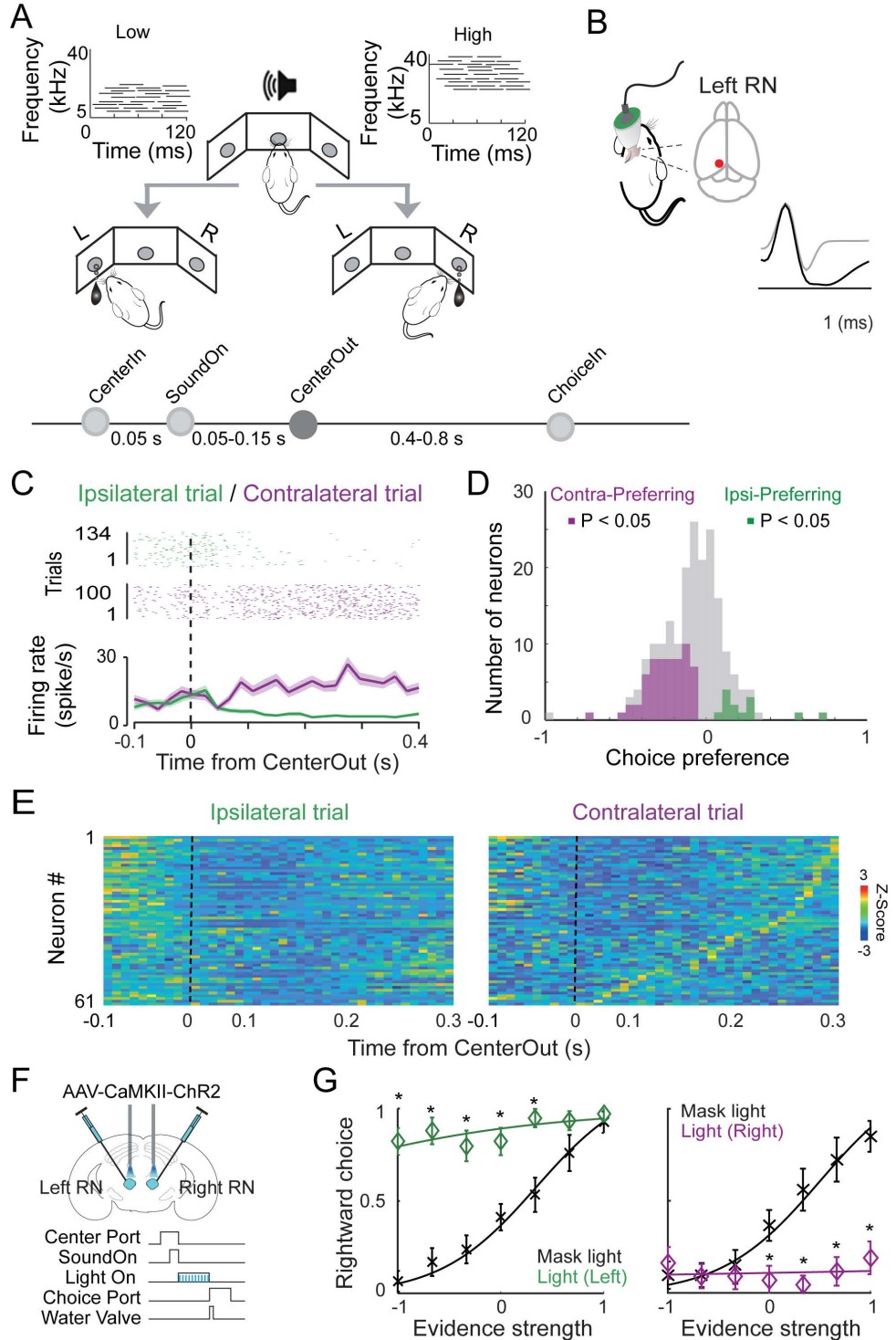

**Fig 1. RN excitatory neurons are preferentially activated during the contralateral movements in an auditory task.**
(**A**) Schematic of the auditory frequency discrimination task. In this two-alternative forced-choice task, mice poke the center port to trigger auditory cues. Mice should then poke one of the side ports for water reward depending on the tone frequencies (low frequency to left side, high frequency to right side). (**B**) Left: schematic of tetrode recording in the RN of a mouse. Right: example waveforms of excitatory (black) and inhibitory (grey) units. (**C**) One example RN excitatory neuron with contralateral preference. Top: Raster plots of neuronal activity for individual ipsilateral (green) and contralateral (purple) trials aligned to the time when the mouse withdrew from the center port (Center-Out). Bottom: PSTH for ipsilateral (green) and contralateral (purple) trials. (**D**) Histogram plots show the number of

neurons (198 units total) with contralateral preference (purple, 61 units) and ipsilateral preference (green, 13 units). $\chi^2$ stat = 38.2904; p = 6.096e – 10. (**E**) Heatmap showing the activity of neurons with contralateral preference (Wilcoxon signed-rank test, p < 0.05, n = 61 neurons from 5 mice). Color-coded neural traces were normalized to the peak of each cell's mean firing rate on the contralateral trial (purple) and sorted by the peak time. 61/198 with contralateral preference vs. 13/198 with ipsilateral preference; $\chi^2$ stat = 38.2904; p = 6.096e – 10). (**F**) Schematic of viral delivery and photoactivation of either left or right RN during the task. 5 ms light pulses at 20 Hz for the whole duration of indicated time window. (**G**) Left: psychometric curves of mice' performance in control (black) and photostimulation of left RN (green) trials. Right: psychometric curves of mice' performance in control (black) and photostimulation of right RN (purple) trials. For the left RN, *n* = 16 session from 3 mice (5 sessions for mouse 1, 3 for mouse 2, 8 for mouse 3), *p* = 4.3778e – 04; for the right RN, *n* = 16 session from 3 mice (7 sessions for mouse 1, 4 for mouse 2, 5 for mouse 3), *p* = 6.1702e – 07, error bars, mean ± s.e.m. Wilcoxon sign-rank test. Underlying data can be found in the S1 Data.

expressing channel rhodopsin 2 (ChR2) [20] under the CaMKII promoter, termed as AAV-CaMKII-ChR2, into the RN bilaterally. Optic fibers were then implanted right above the RN (Fig 1F). We found that activation of the RN excitatory neurons when mice moved from the center port to a side port induced a substantial contralateral bias (Fig 1G). To rule out the possibility that light stimulation itself caused an effect on the biased behaviors, we expressed green fluorescence protein (GFP) in the left RN and performed the same set of tests with optical stimulation. Light stimulation in the RN itself caused no detectable effect on the task performance (S3 Fig).

Altogether the findings indicate that RN neurons are preferentially activated for the contralateral movement in the goal-directed auditory task, and their activity effectively promoted the choice of the contralateral movement.

## The red nucleus receives projection from zona incerta parvalbumin-expressing neurons

To seek neural circuits projecting to the RN that may regulate its activity to impact the movements in this goal-directed auditory task, we used an engineered rabies viral system [21] to retrogradely label brain regions directly projecting to the RN. As illustrated in Fig 2A, to specifically express GFP, TVA, and oG in RN excitatory neurons, we injected AAV-FLEX-TVA-P2A-eGFP-2A-oG into the left RN of adult VGlut2-Cre mice [22]. Three weeks after the AAV injection, we performed the second injection in the same injection site with the EnvA G-deleted Rabies-mCherry, which can infect those RN neurons expressing TVA. GFP and mCherry will both be expressed in RN excitatory neurons, which are defined as starter neurons. The mCherry will retrogradely cross synapses in the presence of oG neurons projecting to the starter neurons. We collected the brain tissue and prepared brain slices for confocal imaging one week after the rabies infusion. We detected mCherry+ neurons in a group of brain regions, including the motor cortex, substantial nigra pars reticulata, subthalamic nucleus and zona incerta (ZI) (Fig 2B).

Collective evidence has indicated the ZI functions in controlling visceral activity, attention, arousal, posture or locomotion, eating, hunting, defensive and nocifensive behaviors, and potentially encodes actions [10–19]. Clinically, the ZI has been recently suggested as a candidate for deep brain stimulation to treat Parkinson's Disease [23]. Given its potential role in motor functions, the ZI has been proposed to be an integrative node for generating direct responses to a given sensory stimulus [11,24]. We therefore determined the possible role of the ZI in regulating the activity of the RN in auditory cued goal-directed task.

Most of the retrogradely-traced neurons in the ZI were in the ventral portion of the ventral medial ZI (vZIm) (Fig 2B). The majority of ZI neurons express GAD, an interneurons marker [11]. To verify the interneuron types in the vZIm, we first immunostained the ZI brain slices

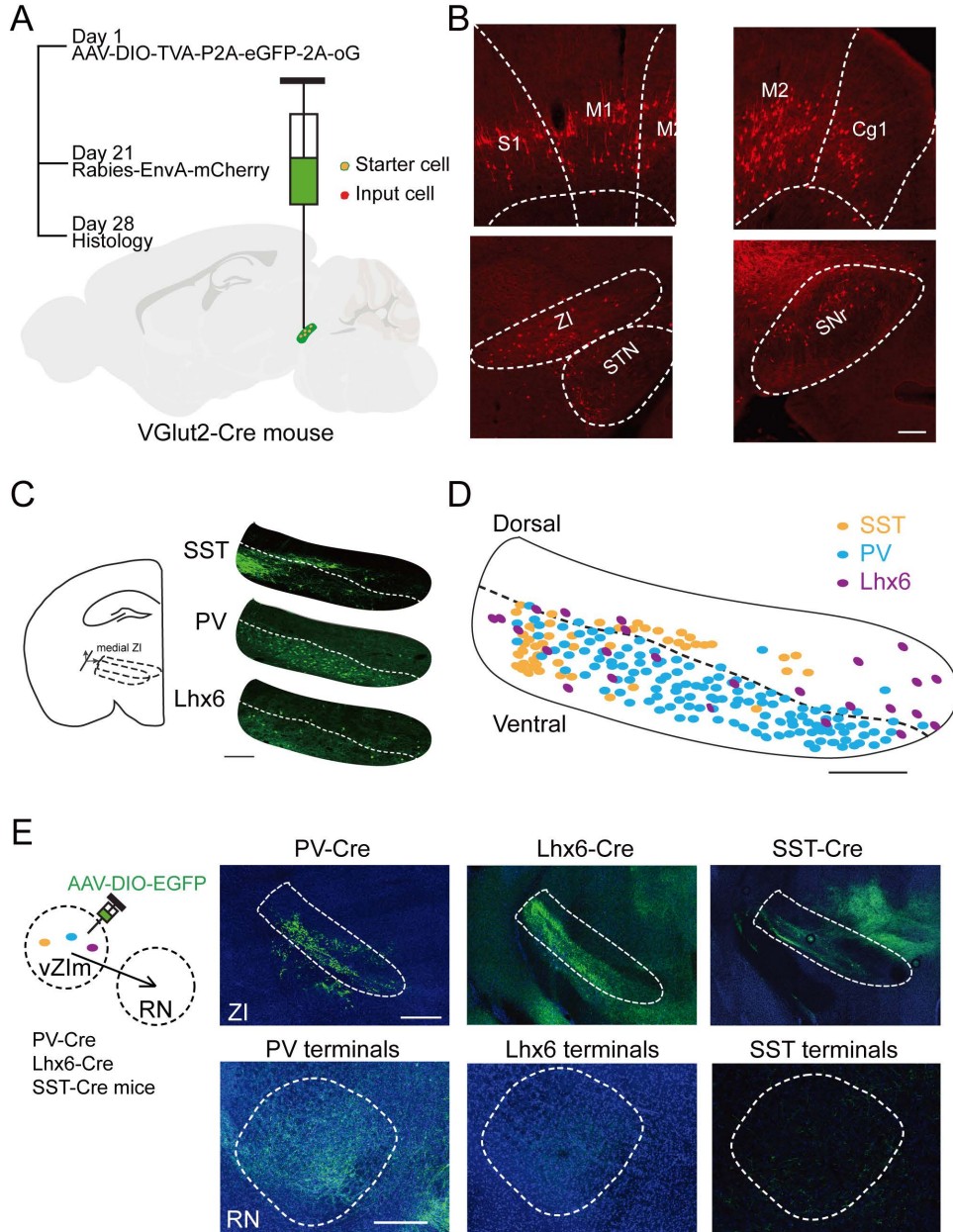

**Fig 2. RN excitatory neurons receive ZI PV neuron projection.** (**A**) Schematic of monosynaptic retrograde tracing from RN excitatory neurons from 3 mice. (**B**) Representative mCherry+ neurons in motor cortex (M1, M2), Cingulate cortex (Cg1), Somatosensory cortex (S1), Subthalamic nucleus (STN), Substantia nigra pars reticulata (SNr) and Zona incerta (ZI), that directly project to RN excitatory neurons. (**C**) Left: Schematic of anatomical location of medial ZI; Right: Immunostaining for PV, Lhx6 and SST cells in the medial ZI. The dash line indicates the border between the dorsal and ventral part of the ZI. Scale bar: 200 μm. (**D**) Reconstruction of PV (blue), Lhx6 (purple) and SST (orange) cells in the medial ZI. Dash line indicates the border between the dorsal and ventral part of the ZI. Scale bar: 200 μm. (**E**) ZI projections to the RN from PV-Cre (left), Lhx6-Cre (middle), and SST-Cre (right) mice. Upper row shows the representative images from injection sites (ZI), lower row shows from the projection sites (RN). Underlying data can be found in the S1 Data.

with different types of interneuron markers. There were three types of inhibitory neurons, parvalbumin (PV)-, somatostatin (SST)-, and LIM homeobox protein 6 (Lhx6)-positive neurons (Fig 2C and 2D), consistent with previous findings [11].

We next examined whether all these types of interneurons project to the RN. AAV-DIO-GFP, with a Cre-dependent expression of GFP cassette, was infused into the left ZI of PV-Cre, Lhx6-Cre or SST-Cre mice, respectively (Fig 2E). Three weeks after the viral infusion, we collected the brain tissue and analyzed the GFP signal in the RN by confocal imaging method. We found that there were GFP+ terminals in the ipsilateral RN (indicating projections from the ZI) only from the PV-Cre mice. In the Lhx6- and SST-Cre mice, we did not detect GFP+ signal in the RN (Fig 2E). Together our results indicate that ZI PV neurons, but not Lhx6 and SST neurons, project to the RN.

### Activation of parvalbumin-expressing neurons in the zona incerta inhibited red nucleus neurons and biased the movement choices in the auditory task

The projection of PV neurons in the ZI to the ipsilateral RN motivated us to explore its potential role in regulating the RN's neuronal activity and function in the auditory task. First, we optogenetically activated ZI PV terminals in the RN and recorded RN neuronal activity. As illustrated in Fig 3A, we expressed ChR2 in PV neurons of the left ZI by injecting an AAV carrying DIO-ChR2, a Cre-dependent expression cassette of ChR2, into left ZI of PV-Cre mouse. A cannula was implanted right above the left ZI. A tetrode bundle integrated with an optical fiber, termed optrode, was implanted in the left RN. Three weeks after the surgery, neuronal activity of the left RN upon optical stimulation was recorded using the tetrode method as previously described [2,3]. To prevent the interference of antidromic activation of ZI PV soma, we infused TTX (1 μM, 300 nl) through the cannula in the ZI during the recording. We compared the firing rates between baseline (50 ms before the light onset) and light period (0–500 ms) for each recorded neuron, using Wilcoxon rank-sum test. Among the 44 identified units, 23 of them dramatically decreased firing rates in response to the light stimulation, 3 of them increased the firing rates and the rest 18 units showed no detectable changes in firing rates (Fig 3A and 3B). Overall, it indicates that ZI PV neuronal activation suppressed neuronal activity in the RN.

We next explored whether such inhibition by ZI PV neuronal activation to the RN affects the movements to reward ports in the task. In a cohort of adult PV-Cre mice, after being well-trained in performing the auditory task, we injected the AAV-DIO-ChR2 into the ZI bilaterally, followed with fiber cannula implantation. Three weeks after the surgery, we optogenetically activated the ChR2 expressing PV neurons either in the left side or right side of the ZI during the movement period. We found that the optical activation of PV neurons of the ZI substantially biased the movements to ipsilateral port (Fig 3C and 3D). Interestingly, optical activation of ZI PV neurons only during the first 200 ms of movement, but not the later phase, induced ipsilateral bias (S4A and S4B Fig).

To exclude any potential effect from optical stimulation, we performed a similar set of tests by injecting AAV-DIO-GFP, in which optical stimulations exhibited no detected effect on the movement (S4C Fig). To rule out the possibility of ZI PV neuron collateral activation effect, we repeated the above behavioral test from a different cohort of mice in the following condition: (1) infused TTX (1 μM, 300 nl) into the right-side of ZI through an implanted cannula, and (2) photo-stimulating ZI PV neuronal terminals in the RN through optic fibers implanted in the right-side of RN (Fig 3F, left panel). Under this condition, photo-stimulation is specific to the ZI PV to the RN projection and there should not be antidromic activation of ZI PV soma. We found that this manipulation induced rightward bias in mouse behavior (Fig 3F, right panel), indicating that ZI PV projection to the RN could modulate movements directed by auditory cue.

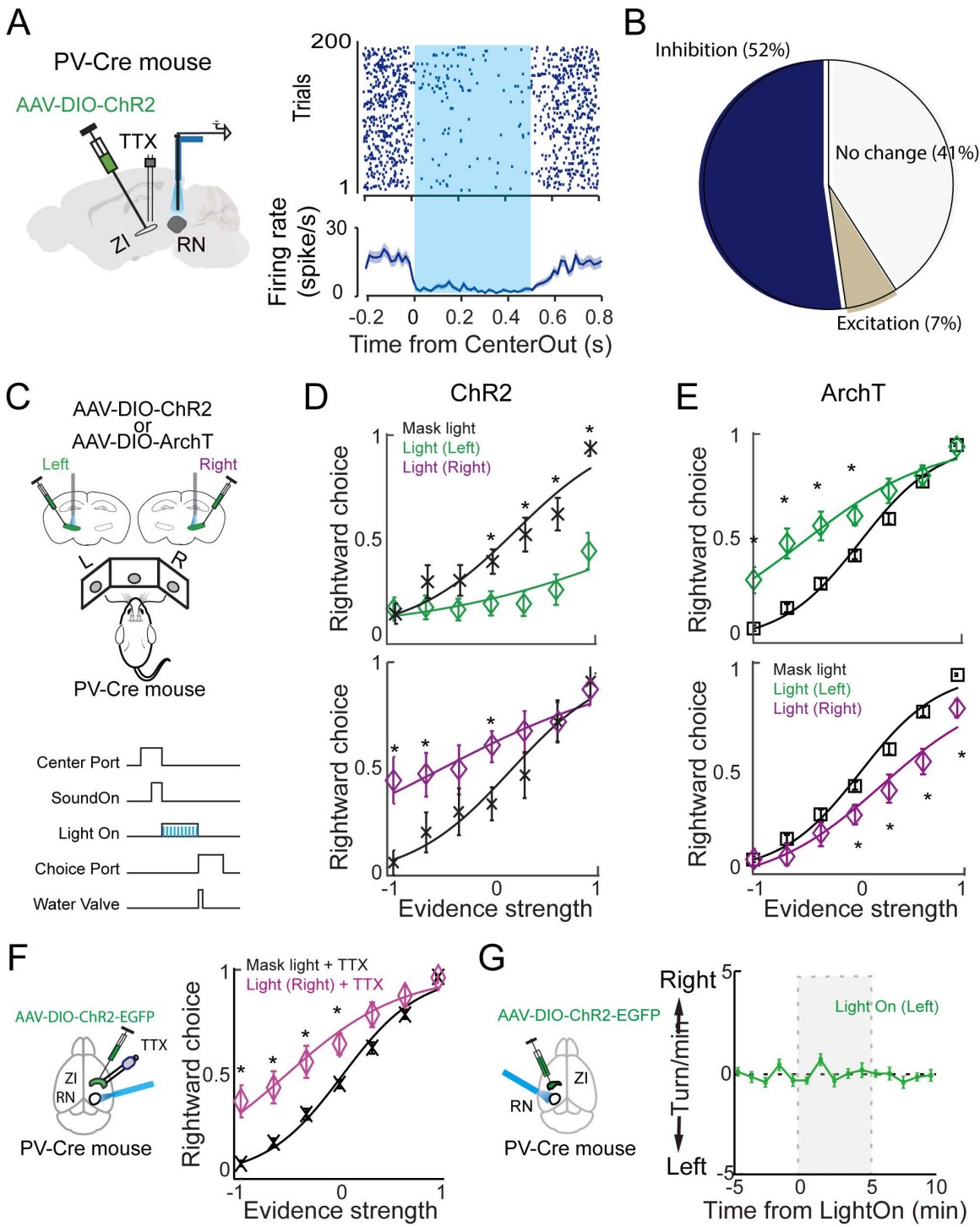

**Fig 3. ZI PV neurons inhibited RN neuronal activity and biased mice's performance.** (**A**) Left: Schematic of photostimulation and tetrode recording under the local infusion of TTX into the ZI. Right: representative raster plot and PSTH plot from one recorded unit. (**B**) A pie chart showing the changes in RN neuronal activity upon ZI PV neuronal terminal opto-stimulation and TTX infusion simultaneously. (**C**) The experimental design to photostimulate or photosilence ZI PV neurons during task performance. Upper panel: Vial delivery and photo manipulations of PV cells in the ZI. Lower panel: Photo manipulation time window (blue bars, movement epoch). (**D**) Psychometric curves of mice' performance in control (black) and photostimulation of left ZI (green) trials or right ZI (purple) trials. *n* = 19 sessions from 6 mice (2 sessions for mouse 1, 4 for mouse 2, 2 for mouse 3, 3 sessions for mouse 4, 4 for mouse 5, 4 for mouse 6), error bars, mean ± s.e.m. *p* = 0.0017; Wilcoxon sign-rank test. (**E**) Psychometric curves of mice' performance in control (black) and photosilencing of left ZI (green) trials or right ZI (purple) trials. *n* = 25 session from 4 mice (7 sessions for mouse 1, 4 for

mouse 2, 5 for mouse 3, 9 for mouse 4), error bars, mean ± s.e.m. $p = 2.e - 05$, Wilcoxon sign-rank test has been used for statistical analysis. (**F**) Left: Schematic of combining TTX fusion into ZI with RN terminal photostimulation. Right: Psychometric curves of mice' performance in control (black) and photostimulation of right RN + TTX (purple) trials. $n = 26$ session from 4 mice (5 from mouse 1, 7 from mouse 2, 8 from mouse 3, 6 from mouse 4), error bars, mean ± s.e.m. $p = 0.0186$, Wilcoxon sign-rank test. (**G**) Photostimulation of ZI PV terminals in the RN did not change mouse's movement directions out of the behavioral task. mean ± s.e.m. 42 trials from 3 mice. Underlying data can be found in the S1 Data.

To further validate this observation, in another cohort of adult PV-Cre mice, we infused AAV-DIO-ArchT, a Cre-dependent expression cassette of ArchT, an inhibitory optogenetic gene [25], into the ZI. The experimental procedures were performed as those of ChR2 tests. As expected, when we optogenetically inhibited the PV neurons during the movement period, the performance was biased to the contralateral port (Fig 3E). This is consistent with the findings when we activated the RN neurons (Fig 1F and 1G).

To assess whether the observed behavioral effect is task related, we performed a similar ZI PV terminal photo-stimulation in mice that are not performing the task. We found that stimulating ChR2+ ZI PV terminals in left RN induced no clear directional movements (Fig 3G), indicating that the ZI PV to the RN projection is likely to modulate goal-directed movements.

It has been shown before that PV and Lhx6 neurons in external globus pallidus contribute oppositely to motor suppression [26]. The substantial impact of activation or inhibition of ZI PV neurons projecting to the RN on the task movement motivated us to examine the other population of ZI interneurons. We then determined whether the Lhx6+ neurons in ZI are involved in regulating the task movements. Similar to the experiments on testing the ZI PV neurons, we used Lhx6-Cre mice for optogenetic manipulations. We found that activation of Lhx6 neurons caused a dramatic contralateral bias. In contrast, in a cohort of SST-Cre mice, we found that activation of vZIm SST neurons showed no detectable effect on the task movement (S4D–S4G Fig). Note that the biased effect of LhX6 activation is opposite to that of PV neuronal activation (Fig 3C).

To further understand how ZI PV and Lhx6 neurons are involved in the movements, we used a ChR2-assisting tagging approach [27] to record ZI PV or Lhx6 neurons from mice performing the task. Similar to the strategy described above, we specifically express ChR2 in ZI PV (or Lhx6) neurons using PV-Cre (or Lhx6-Cre) mice. These neurons can be activated by blue light pulses with short latencies and follow high-frequency light stimulations (S5A– S5C Fig). We found that 43% identified PV neurons ($n = 30$ neurons) displayed contralateral movement preference, 7% displayed ipsilateral movement preference, and the rest with no preference (S5D Fig). On the contrast, 42.9% identified Lhx6 neurons ($n = 8$ neurons) displayed ipsilateral movement preference, 14.2% displayed contralateral preference, and the rest with no preference (S5E Fig). All the identified SST neurons ($n = 11$) displayed no preference (S5F Fig). Together, these results indicated that PV and Lhx6, but not SST neurons in the vZIm are preferentially activated in contralateral and ipsilateral movements directed by auditory decisions, respectively. This is consistent with optogenetic manipulation results in Fig 3.

Altogether, using optogenetic activation of individual populations of ZI interneurons we showed that the ZI via its PV neurons suppress RN activity and dampens the contralateral movement in the auditory-cued goal-directed task. Unexpectedly, we found that Lhx6 neurons although not project to the RN their activation motivates a contralateral movement.

## Lhx6-expressing neurons inhibit PV-expressing neurons in the zona incerta

The findings that ZI Lhx6 neurons do not directly project to the RN (Fig 2E) and activations of ZI PV and Lhx6 neurons caused opposite bias directions in task movements (Figs 3 and

S4), led us to determine the possibility that ZI Lhx6 neurons regulate task movement through inhibiting ZI PV neurons locally.

To test this hypothesis, we expressed ChR2 in ZI Lhx6 neurons and recorded light-evoked neuronal activity in the ZI using tetrode recording method (Fig 4A). Similar to the optogenetic tagging method with optrode as described previously [27], neurons activated by light pulses with short latencies and also able to follow with high-frequency light stimulations were considered as ChR2 expressing Lhx6 neurons (see section "Materials and methods"). Out of the total 72 detected units, we found 2 units were activated by the light pulses and followed the 20 Hz light stimulation (Fig 4B, upper panel), thus they were defined as ChR2+ neurons. The activity of units that were inhibited or unchanged by the light pulses are defined as ChR2– neurons (Fig 4B, lower panel). Among these 70 ChR2– neurons, 54 units showed strong light-evoked inhibitory effects (Fig 4C and 4D). Because most inhibitory neurons in the vZIm are PV expressing neurons (Fig 2D), we expected most of the 54 units were PV expressing neurons, suggesting that Lhx6 neuron activation likely suppressed PV neuron activity.

To examine whether the intra-ZI connections are a common feature of ZI interneurons, we performed the same tests as for Lhx6 on PV neurons. Among the 88 recorded neurons, 8 of them were ChR2+ and 80 were ChR2–. In contrast to the Lhx6 activation experiment, when we activated ChR2 expressing ZI PV neurons, we found no obvious effect on ChR2 negative neurons in the ZI (Fig 4E–4H). Only one neuron showed a potential inhibitory response (Fig 4H, lower panel). To validate the possibility that Lhx6 neurons form robust synapses with local neurons, we performed in vitro slice recordings. We found that the activation of Lhx6 but not PV neurons induced local postsynaptic activation (Fig 4I and 4J), suggesting that Lhx6 neurons form strong intra-ZI synaptic connections. Because most inhibitory neurons in the vZIm are PV expressing neurons (Fig 2D), we speculated that some of the 43 inhibited neurons in Fig 4C and 4D were PV expressing neurons, suggesting that Lhx6 neuron activation likely suppressed PV neuron activity.

Altogether, the findings in Figs 3 and 4 suggest Lhx6 neurons may likely exert their regulation to the task movement via inhibiting ZI PV neurons locally. In this task, PV neurons in the ZI may integrate local activities (i.e., from Lhx6 neurons) and then impact RN activity to regulate the movement to the reward ports, as proposed in Fig 4K.

## Discussion

In this study, we found that the ZI to RN circuit plays an important role in regulating lateral movements directed by auditory cues. RN neurons are activated preferentially during contra-lateral movements in the task. Optogenetically activating RN neurons biased mice's movements in the task toward the contralateral side. We further revealed that PV neurons in the vZIm project to the RN and importantly counteract the movements directed by the auditory decisions in the task. Altogether, our findings for the first time demonstrate an inhibitory function of the ZI-RN in regulating auditory-cued goal-directed movements.

Many brain regions have been implicated in driving sensory-cued goal-directed movements. To understand how auditory decision links to motor execution, we examine the potential role of the RN in a two-alternative force choice auditory discrimination behavior. The RN is integral to the execution and coordination of goal-directed movements, functioning through its connections with the cerebellum, motor cortex, and spinal cord [28]. Its role in fine motor control and integration of sensory feedback underscores its importance in precise and adaptive motor behaviors [9]. RN neurons display target location-dependent modulation in firing rate during reaching to grasp [29], and the directional movement signals in RN neurons are increased in trials when animals exhibited cognitive control [30]. In our behavioral task, the longest movement period within a trial is when the mice withdrew from the

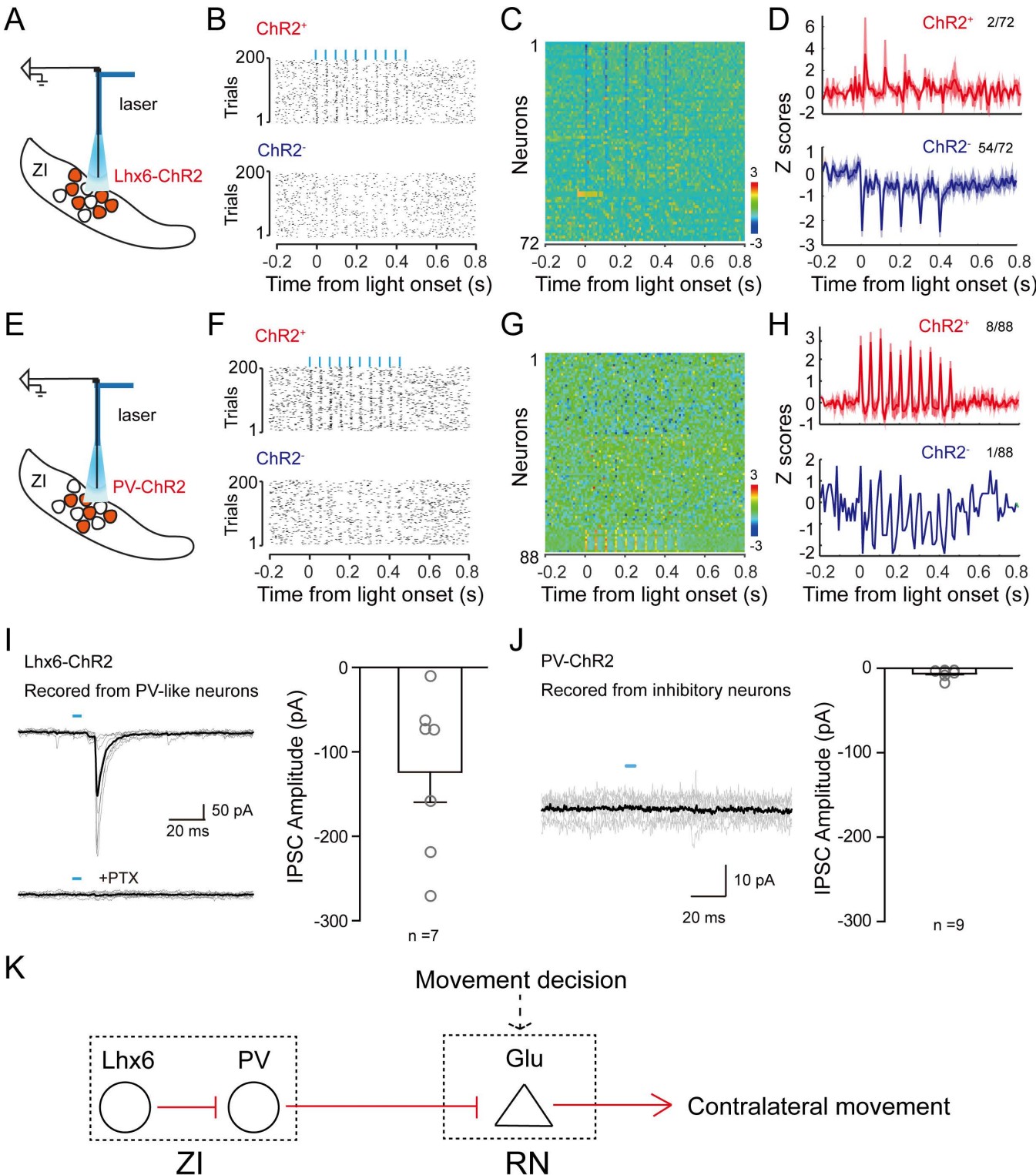

**Fig 4. ZI-Lhx6 neurons inhibited other ZI neurons.** (**A**) Schematic of tetrode recording in ZI where Lhx6 neurons express ChR2. (**B**) Raster plots show two neurons' responses to photostimulation. Upper panel: ChR2+ neuron; lower panel: ChR2– neuron. 5 ms light pulses for 500 ms at 20 Hz. (**C**) All recorded neurons from the experiment illustrated in (A). Color-coded neural traces were normalized to the peak of each cell's mean firing rate and sorted by modulation index during the light stimulation. (**D**) Average *z*-score of ChR2+ neurons (red, *n* = 2 of 72 across 4 mice), and inhibited ChR2– neurons (blue, *n* = 54 of 72) during photostimulation. 2/72 with activation vs. 54/72 with silencing; χ² stat = 54.3354; *p* = 1.6909*e* – 13. (**E**) Schematic of tetrode recording in ZI where PV neurons express ChR2. (**F**) Raster plots show two neurons' responses to photostimulation. Upper panel: ChR2+ neuron; lower panel: ChR2– neuron.

500 ms light stimulation at 20 Hz. (**G**) All recorded neurons from the experiment illustrated in (**E**). Color-coded neural traces were normalized to the peak of each cell's mean firing rate and sorted by modulation index during the light stimulation. (**H**) Average $z$-score of ChR2+ neurons (red, $n = 8$ of 88 across 4 mice), and inhibited ChR2– neurons (blue, $n = 1$ of 88) during photostimulation. 8/88 with activation vs. 1/88 with silencing; $\chi^2$ stat = 5.7379; p = 0.0166. (**I**) Left: An example of whole-cell patch recording from ZI inhibitory neurons where Lhx6 neurons express ChR2. Blue tick, 5 ms photo stimulus to activate ChR2. PTX: picrotoxin applied through bath solution. Right: Population summary of inhibitory postsynaptic current (IPSC) amplitude ($n = 7$). (**J**) Left: An example of whole-cell patch recording from ZI inhibitory neurons where PV neurons express ChR2. Blue tick, 5 ms photo stimulus to activate ChR2. Right: Population summary of inhibitory postsynaptic current (IPSC) amplitude ($n = 9$). (**K**) Proposed working model of ZI-RN circuit in controlling contralateral movements. Underlying data can be found in the S1 Data.

center port and moved towards the side ports for water reward. We found a large portion of RN neurons increased their firing rates during this period and optogenetically activating them biased the mice's movements towards contralateral side (Fig 1), supporting the notion that the RN controls the movements directed by auditory decisions in the task. How does the auditory decision reach to the RN remain unknown. Our retrograding tracing from the RN (Fig 2) identified substantial nigra pas reticulata (SNr) as one of the direct upstream brain regions. Since SNr receives direct inputs from the D1 neurons in the dorsal striatum including the tail striatum, and D1 pathway was shown to drive the performance in this task [4], SNr may be the locus linking the RN to the auditory decisions. Other identified regions that directly project to the RN include motor cortex (M1 and M2) and subthalamic nucleus, which may also be involved in directing the movement signals to the RN in this task.

About 85% neurons in the ZI are GABAergic with differential molecular markers and sectional distribution [11]. Based on our tracing results (Fig 2) we found the PV neurons in vZIm directly project to the RN. Through inhibitory input, PV neurons in vZIm counteract the movements directed by the auditory decisions (Fig 3). Interestingly, ZI PV manipulation during the task was only effective when it started at the beginning of the movement (S4A and S4B Fig). Photo-activating ZI PV neurons 200 ms after the movement initiation has no effect on mice's performance (S4A and S4B Fig). This result suggests that ZI PV modulation on the RN is to the movement decision and/or coordination, rather than directly induce the directional movement. Furthermore, photo-activation of ZI PV neuronal terminals in the RN outside the task induced no specific directional movement, indicating its task-specific involvement. Combined with the previous reports of RN function in voluntary movements (discussed in the above paragraph), our findings support the notion that ZI PV to the RN pathway modulates goal-directed movements.

It is not clear why the brain would have this counteracting circuit during this goal-directed movement. One possibility is that ZI-RN may function as a balance control of the body posture during the movements, since ZI has been reported to control posture [31,32]. Where does the ZI receive the driving signals for this activity during the movement? How does the ZI communicate with the RN during the movement? More experiments with analyses of detailed movement stages and ZI/RN activities are needed to address these questions.

In our study, the reward sides are cued by the auditory signals (Fig 1A). It is highly possible that the RN and ZI receive direct or indirect inputs from other sensory modalities and function similarly in other sensory-cued goal-directed movements. Future experiments with other sensory cues are necessary to address whether our findings are generalized across different sensory modalities.

## Materials and methods

### Animals

Animal procedures were approved by the Stony Brook University Animal Care and Use Committee and carried out in accordance with the National Institutes of Health standards

(approval number: 824,397). Male and female C57BL/6J mice (Charles River), PV^Cre (JAX strain #017320), SOM^Cre (JAX strain #013044), Lhx6^Cre ([33]), and GAD^Cre (JAX strain #010802) were housed with free access to food, but water restricted after the start of behavioral training. During training, water was available based on task performance (2.5 µl for each correct trial); whereas during non-training days water bottles were provided to the mice for at least 1 h per day.

## Viral injection

Mice aged 6–8 weeks were anaesthetized with 1–2% isoflurane and placed in a stereotaxic apparatus. Viral injections were performed using previously described procedures at the following stereotaxic coordinates: ZI: 2.0 mm caudal from bregma, 1.6 mm lateral from midline, and 4.0 mm depth from cortical surface; RN: 3.6 mm caudal from bregma, 0.6 mm lateral from midline, and 3.5 mm depth from cortical surface. A small craniotomy was made according to the coordinates, and a home-made glass micropipette (tip diameter of 10–15 µm) was inserted from the surface of the brain. The virus was delivered through the glass pipettes that were connected to a Picospritzer II microinjection system (Parker Hannifin Corporation). For cell-type-specific viral infection of ZI neurons, we injected 200–500 nL of AAV5-EF1α-DIO-ChR2-eGFP (University of North Carolina Vector Core, Chapel Hill, NC). Injection speed was 100 nL min⁻¹ and the injection needle was raised 10 mins after completion. Fourteen days (for AAV) or after injection, mice were perfused, and brain slices were collected for imaging. Images were acquired with a laser-scanning confocal microscope (FV1000, Olympus).

For activation of the ZI PV, SST, GAD and Lhx6 population, we injected 200–500 nL AAV5-EF1α-DIO-ChR2-eGFP bilaterally. Manipulation experiments were conducted 4 weeks after the viral injections.

For characterization of inputs to specific ZI populations using rabies tracing, PV^Cre or Lhx-6^Cre mice were injected in the ZI (coordinates above) with a mixture of DIO-TVA-mCherry and rabies glycoprotein (AAV5-CAG-FLEX-OG-TVA-mCherry) (Salk Institute). On day 21, mice were injected with EnvA-pseudotyped G-deleted rabies-eYFP, which only infects cells expressing the TVA receptor. Tissue was fixed for analysis at day 30 and brain slices were collected for imaging. Images were acquired with a laser-scanning confocal microscope (FV1000, Olympus).

## Behavioral training

The mice were placed on a water deprivation schedule and trained to perform an auditory 2AFC task in a single-walled sound-attenuating training chamber as described previously. Behavioral system is controlled, and data is analyzed by Bpod system (Sanworks, https://sanworks.io/ provides all source files). In brief, freely moving mice were trained to initiate a trial by poking into the center port of a three-port operant chamber, which triggered the presentation of a stimulus. Subjects then selected the left or right goal port. The cloud-of-tones stimulus consisted of a stream of 30 ms overlapping pure tones presented at 100 Hz. The stream of tones continued until the mouse withdrew from the center port. Eighteen possible tone frequencies were logarithmically spaced from 5 to 40 kHz. For each trial, either the low stimulus (5–10 kHz) or high stimulus (20–40 kHz) was selected as the target stimulus, and the mice were trained to report low or high by choosing the correct side of port for water reward. Correct responses were rewarded with water (2.5 µl for each correct trial), and error trials were punished with a 4 s timeout. Sound intensity was calibrated at 65 dB SPL. The evidence strength $r$ determined the difference in the rate of

high and low octave tone in the stimulus. Tones were drawn from the target octave with a probability of $1 + 2r/100/3$. To quantify mice's performance in the task, we used a logistic regression model described before [2,3,6]. $\log(p/(1 - p))$ $\beta_0 + r*\beta_1$, where $p$ is the fraction of choices towards the port associated with high frequencies. Parameters $\beta_0$ and $\beta_1$ measure the bias and slope of the psychometric curve. Reaction time was calculated as the time between the onset of tone and the time of withdraw from the center port. Movement time was calculated as the time between the time of withdraw from the center port and the onset of poking the side port.

## Tetrode recording

Custom tetrode and optic fiber arrays were assembled in house. Each array carried 8 tetrodes and one optic fiber (62.5 μm diameter with a 50 μm core; Polymicro Technologies). Each tetrode consists of 4 polyimide-coated nichrome twisted together and gold-plated to an impedance of 0.3–0.5 MΩ at 1 kHz (wire diameter of 12.7 μm; Sandvik in Palm Coast, FL). The fiber tips were sharpened at the points using a diamond wheel to improve tissue penetration and increase the light illuminating area. The resulted optrodes were mounted on vertically movable microdrive. The optrode tips were coated with DiI to assist the identification of fiber tracks in brain tissues. To implant the optrode array, mice were anaesthetized with 1–1.5% isoflurane and placed in a stereotaxic apparatus (Kopf). A craniotomy was made over the target area. The dura was removed and the implant was placed over the target area and fixed in place with dental acrylic. The tetrode was then lowered down to the ZI or RN with close recording monitoring (75 μm maximum per day).

Electrical signals in ZI or RN were recorded using Neuralynx Cheetah 32-channel hybrid system and cheetah data acquisition software. Signals were filtered 600–6,000 Hz. Single units were isolated offline using Spike3D and MClust3.5. Clusters with isolation distance >20 and L-Ratio < 0.1 were included.

For all the isolated single units, those having waveforms with half-volley-width less than 100 μs are identified as interneurons, those having waveforms with half-volley-width more than 150 μs are identified as excitatory neurons as described previously.

## Opto-genetic manipulation

For opto-genetic manipulation of the different population in the ZI, well-trained mice were bilaterally injected with the AAV5-EF1α-DIO-ChR2-eGFP or AAV5-EF1α-DIO-ArchT-eGFP (UNC vector core) into the ZI. For TTX infusion, the guide cannula (26 Gauge, 0.46 mm OD, RWD) was lowered into the ZI. TTX (1 μM, 300 nl) was infused into the ZI before the behavioral test or multi-unit recording. The optic fibers were inserted into the ZI or RN of the mice. Laser light was adjusted to produce the desired output at the end of the patch cord. For ChR2, 480 nm (5 ms pulses at 20 Hz), or for ArchT 530 nm (continuous) laser light was generated from a diode-pumped solid-state laser (Shanghai Dream Lasers, Shanghai, China) and couple to the optic fiber through an FC/PC patch cord using a FiberPort Collimator (Thor Labs), yielding an average power of 5 mW at the optic fiber tip outside the tissue. Light pulses were delivered during the different epoch as indicated in the figures. Manipulation trials were randomly interleaved with control trials.

To control for the potential influence of visual stimulation effects, during all trials, we included a masking light stimulation using the same presentation time window and wavelength as photo stimulation trials delivered through a bulb placed above the center port, as previously employed [4,6].

## Optical identification of ChR2-expressing neurons

At the beginning of each experiment involving identified ChR2-expressing neurons, the probe was lowered to the presumed depth of the ZI through the burr hole and allowed to settle for 10–45 mins. The fiber attached to the probe was coupled to a 473 nm diode-pumped solid-state laser (Shanghai Dream Lasers, Shanghai, China) using an optical multimode fiber (200 μm, 0.39 NA FC/PC, Thorlabs part no. M83L01). Once a stable recording was established, blue light was flashed for 10 ms at 2–10 mW through the fiber into the brain at 20 Hz for 200 repetitions. Laser power was adjusted to minimize the latency of activation while also minimizing optical artifacts. To identify neurons with ChR2-mediated responses, we performed the following three analyses as previously reported [27,34]: (1) distribution of spiking timing; (2) Latency; (3) spike waveform correlation. A unit (neuron) was determined as ChR2-expressing if: (1) we can detect a reliable increase of spike precisely time-locked to the onset of light pulses at 20 Hz, (2) the light response has a latency less than 10 ms, and (3) the spike waveforms have correlation of more than 0.95. Once a neuron was identified as possibly being light active, the locomotor/stimulation session would proceed after which a post identification session would be carried out to ensure another unit had not moved into the recording space. Final clustering was performed post hoc.

## Preference analyses

To assess the significance of choice preference, we first compared mean firing rates between correct left and right trial during the different phase (bin size = 10 ms, time window: −100 to 300 s (CenterOut)) using Wilcoxon sign-rank test. We used a criteria $p < 0.05$ to determine significance. To quantify the selectivity of single neurons for task variables (direction of movement), we used an algorithm based on receiver operating characteristic (ROC) analysis. This analysis calculates the ability of an ideal observer to classify whether a given spike rate was recorded in one of two conditions (e.g., during leftward or rightward movement). We defined "preference" as $2*(\text{ROC area} - 0.5)$, a measure ranging from −1 to 1, where −1 signifies the strongest possible preference for one alternative, 1 signifies the strongest possible preference for the other alternative, and 0 signifies no preference. This analysis was used to test left versus right direction preference (left = −1, right = 1). Statistical significance was determined with a permutation test: we recalculated the preference after randomly reassigning all firing rates to either of the two groups arbitrarily, repeated this procedure 1,000 times to obtain a distribution of values, and calculated the fraction of random values exceeding the actual value. For all analyses, we tested for significance at $\alpha = 0.05$. Only neurons with a minimum number of four trials for each analyzed condition and a firing rate above 2 spikes/s for either of the analyzed conditions were included in the analysis. For analyses based on movement from the odor port to reward port, trials in which the movement time was >1.5 s were excluded.

## Supporting information

**S1 Fig. RN inhibitory neuronal activity.** A. Heatmaps show the activity of RN inhibitory neurons (n == 27 neurons from 5 mice) during ipsilateral and contralateral movements in the task. Dash lines indicate the activity alignment to CenterOut. B. Histogram plots show the number of neurons with contralateral preference (purple) and ipsilateral preference (green). Chi2stat == 38.2904; p == 0.684. C. One example RN excitatory neuron with contralateral preference. Top: Raster plots of neuronal activity for individual ipsilateral (green) and contralateral (purple) trials aligned to the time when the mouse withdrew from the center port (CenterOut). Bottom: PSTH for ipsilateral (green) and contralateral (purple) trials. D. One example RN excitatory neuron with contralateral preference during the correct trial (solid

line) and error trial (dash line), ipsilateral (green) and contralateral (purple) trials. Underlying data can be found in the S1 Data.
(TIF)

**S2 Fig. RN excitatory neuronal response to auditory cues.** A. One example RN excitatory neuron during correct (left) and error (right) trials. Raster plots and PSTH plots of neuronal activity for individual low-tone trials (green) and high-tone trials (purple) aligned to sound onset. B. The same example RN excitatory neuron activity aligned to ChoicePortIn. Underlying data can be found in the S1 Data.
(TIF)

**S3 Fig. Control for RN photostimulation.** Left, schematic of viral delivery (AAV-GFP) into the left RN of a wild-type mouse, and photostimulation during the task to rule out light artificial effects on performance. Right, Psychometric curves of mice' performance in control (black) and photostimulation of left RN (green) trials. N == 16 sessions from 3 mice; $p ==$ 0.3045; error bars, mean ±± s.e.m. Wilcoxon sign-rank test. Underlying data can be found in the S1 Data.
(TIF)

**S4 Fig. ZI photostimulation.** A. Psychometric function changes following unilateral photo activation (left hemisphere: green) of PV during the early action epoch (from CenterOut to 200 ms, $p ==$ 0.0328. Wilcoxon sign-rank test) (left panel) or during the later action epoch (200 ms after the CenterOut to the ChoiceIn, $p ==$ 0.4372. Wilcoxon sign-rank test) (right panel). Error bars, mean ±± s.e.m. B. Summary of group data for ipsilateral bias between Mask light and Light groups from Fig 3 (7 sessions from 3 mice), panel A (6 sessions from 3 mice) and B (7 sessions from 3 mice). C. Psychometric curves of mice' performance in control (black) and photostimulation of left ZI (green) trials from mice with ZI PV neurons express GFP. N == 11 sessions from 4 mice, $p ==$ 0.3045. Wilcoxon sign-rank test. D. Schematic of vial delivery and photo manipulations of Lhx6 cells in the ZI. Right, photo manipulation time window (blue bars, movement epoch). E. Psychometric curves of mice' performance in control (black) and photostimulation of left ZI Lhx6 (green) trials or right ZI Lhx6 (purple) trials. Left, n == 19 session from 6 mice, $p ==$ 0.0049; Right, n == 19 session from 6 mice, $p ==$ 0.0132; Wilcoxon sign-rank test. F. Schematic of vial delivery and photo manipulations of SST cells in the ZI. Right, photo manipulation time window (blue bars, movement epoch). G. Psychometric curves of mice' performance in control (black) and photostimulation of left ZI SST (green) trials or right ZI SST (purple) trials. Left, n == 21 sessions from 4 mice, $p ==$ 0.4385; Right, n == 27 sessions from 5 mice, $p ==$ 0.3562; Wilcoxon sign-rank test. Underlying data can be found in the S1 Data.
(TIF)

**S5 Fig. ZI cell-type-dependent neuronal activity during the task.** A. Coronal section from PV-Cre mice (green, PV; red, ChR2). Cre-dependent AAV-ChR2-mCherry were injected into the medial ZI in the PV-Cre mice. Arrows indicate overlapping cells with PV and ChR2 expression. Scale bar, 200 μm. B. Example recording from an identified PV neuron following 10 or 20 Hz light stimuli. C. Opto-tagging verification of PV cells (30 neurons from 6 mice): laser evoked spiking at short latencies (<< 10 ms). D. Top: Example PV cell (raster plot and PSTH plot) shows a significant contralateral preferring during action execution. Dash line is a single trial aligned to CenterOut. Bottom: Average z-score for 43% PV population (n == 30) with contralateral preferring (purple), 7% with ipsilateral preferring (green), and 50% without significant change (black). E. Top: Example Lhx6 cell (raster and PSTH plot) shows a significant ipsilateral preferring during action execution. Dash line is a single trial aligned to

CenterOut. Bottom: Average z-score for 42.9% Lhx6 population (n == 8) with ipsilateral preferring (green), 7% with contralateral preferring (purple) and 50% without significant change (black). F. Top: Example SST cell (raster and PSTH plot) shows a non- significant preferring during action execution. Dash line is a single trial aligned to CenterOut. Bottom: Average z-score for 100% SST population (n == 11) without significant change (black). Underlying data can be found in the S1 Data.
(TIF)

**S6 Fig.  Identification of medial part of ZI and tetrode location.** A. Diagram shows AAV-DIO-EGFP viral expression in medial part of ZI in GAD-Cre mouse. Representative images showing GAD positive cells in ZI from rostral to caudal. B. Left, post hoc identification of tetrode location from the medial ZI. Scale bar, 500 μm. Right, for example single unit waveforms. Scale bar indicates 50 μV and 1 ms. C. Left, post hoc identification of tetrode location from the medial RN. Scale bar, 500 μm. Right, for example single unit waveforms. Scale bar indicates 50 μV and 1 ms. Underlying data can be found in the S1 Data.
(TIF)

**S7 Fig.  RN contralateral preference during the correct and error trials.** A. Heatmap showing a higher neural activity for contralateral preference during correct (upper) and error trials (lower). The color scale represents neural activity, with warm colors indicating higher activity and cool colors indicating lower activity. B. Histogram illustrating the distribution of reaction times across the entire session, with an average reaction time of 0.05 s. C. Histogram depicting the distribution of movement times from the entire session, with an average movement time of 0.8 s. D. Scatter plot showing reaction time from both contralateral trials and ipsilateral trials in Fig 1G (upper) and Fig 3C (lower) in photo-stimulation (green) and control conditions (black). E. Scatter plot showing movement time from both contralateral trials and ipsilateral trials in Fig 1G (upper) and Fig 3C (lower) in photo-stimulation (green) and control conditions (black). Underlying data can be found in the S1 Data.
(TIF)

**S8 Fig.  RN contralateral preference from the trials with different evidence strength.** A. PSTH plots from three example neurons show contralateral preference in both easiest (evidence strength, 1, Left) and the hardest (evidence strength, 0, Right) trials. Neuronal activity for individual ipsilateral trial (green) and contralateral trial (purple) trials was aligned to center port out (CenterOut). B. Quantification of all contralateral preferred neurons under different evidence strengths. n == 61 neurons; evidence strength == 1 Vsversus evidence strength == 0; $p$ == 0.883; paired $t$- test. Underlying data can be found in the S1 Data.
(TIF)

**S1 Data.  All data presented in the current study are organized based on corresponding figure panels.**
(XLSX)

## Acknowledgments

We thank Dr. Gittis' laboratory for the generous gift of the Lhx6-Cre mouse line. We thank Ge and Xiong laboratory members for their valuable comments on the manuscript.

## Author contributions

**Conceptualization:** Liang Chen, Allen P. F. Chen, Qiaojie Xiong.

**Data curation:** Liang Chen, Xinxing Wang, Hanxiao Liu, Chenzhao He, Lu Chen, Thomas A. Kim.

**Formal analysis:** Liang Chen, Xinxing Wang, Hanxiao Liu, Chenzhao He, Lu Chen.

**Funding acquisition:** Qiaojie Xiong.

**Investigation:** Liang Chen.

**Methodology:** Liang Chen, Xinxing Wang, Hanxiao Liu, Allen P. F. Chen.

**Supervision:** Qiaojie Xiong.

**Validation:** Xinxing Wang, Chenzhao He.

**Writing – original draft:** Liang Chen.

**Writing – review & editing:** Liang Chen, Qiaojie Xiong.

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
