## [Editor Report · Decision Letter 0]

8 Jan 2025

Dear Dr Xiong, 

Happy New Year!

Thank you for submitting your manuscript entitled "Zona Incerta negatively regulates the Red Nucleus in an auditory-cued goal-directed movement" for consideration as a Research Article by PLOS Biology.

Your manuscript has now been evaluated by the PLOS Biology editorial staff and I am writing to let you know that we would like to send your submission back for external peer review.

Once your full submission is complete, your paper will undergo a series of checks in preparation for peer review. After your manuscript has passed the checks it will be sent out for review. To provide the metadata for your submission, please Login to Editorial Manager (https://www.editorialmanager.com/pbiology) within two working days, i.e. by Jan 10 2025 11:59PM.

Kind regards,

Christian

Christian Schnell, PhD

Senior Editor

PLOS Biology

cschnell@plos.org

---

## [Decision Letter · Decision Letter 1]

5 Feb 2025

Dear Qiaojie,

Thank you for your patience while we considered your revised manuscript " Zona Incerta negatively regulates the Red Nucleus in an auditory-cued goal-directed movement " for publication as a Research Article at PLOS Biology. This revised version of your manuscript has been evaluated by the PLOS Biology editors, the Academic Editor and two of the original reviewers.

Based on the reviews and on our Academic Editor's assessment of your revision, we are likely to accept this manuscript for publication, provided you satisfactorily address the remaining point raised by Reviewer 1 to state your hypothesis more clearly. Please also make sure to address the following data and other policy-related requests:

* We would like to suggest a different title to improve its accessibility for our broad audience: "The zona incerta negatively regulates the red nucleus during movement cued by sound signals"

* Please edit the abstract (and the entire manuscript) for grammar and clarity. There are a few instances of missing articles, for example it should be *the* zona incerta, and *the* red nucleus everywhere. Red nucleus and zona incerta should also be written without capitalization). 

* Please add the links to the funding agencies in the Financial Disclosure statement in the manuscript details.

* Please include the full name of the IACUC/ethics committee that reviewed and approved the animal care and use protocol/permit/project license. Please also include an approval number.

* DATA POLICY:

Regardless of the method selected, please ensure that you provide the individual numerical values that underlie the summary data displayed in the following figure panels as they are essential for readers to assess your analysis and to reproduce it: 1G, 3DEFG, S3, S4ABCEG, S7DEF and 8B.

* Please ensure that you are using best practice for statistical reporting and data presentation. These are our guidelines https://journals.plos.org/plosbiology/s/best-practices-in-research-reporting#loc-statistical-reporting and a useful resource on data presentation https://journals.plos.org/plosbiology/article?id=10.1371/journal.pbio.1002128

* If you are reporting experiments where n ≤ 5, please plot each individual data point.

* CODE POLICY

We expect to receive your revised manuscript within two weeks. 

*Published Peer Review History*

*Press*

Sincerely,

Christian

Christian Schnell, PhD, 

Senior Editor

cschnell@plos.org

PLOS Biology

Reviewer remarks:

Reviewer #1: The article has improved significantly; the results are now easy to read, and the analysis and experiments are presented logically. However, the authors' path to arrive at the main question remains somewhat elusive. Although the question posed is clear, the hypothesis lacks effective articulation. A well-defined hypothesis should outline a clear cause-and-effect relationship, but unfortunately, this structure is not apparent, leaving the reader grappling with ambiguity.

In this way, in the introduction section, the role of ZI appears abruptly and feels disconnected from the overall narrative. Without adequate context and integration into the hypothesis, the significance of ZI becomes questionable.

In summary, while the article has made notable strides in clarity, improving the articulation of the hypothesis and providing a better context for ZI would greatly enhance the overall coherence and impact of the paper.

Reviewer #2: I was very impressed by the authors' response and edit. All my concerns have been addressed.

---

## [Editor Report · Decision Letter 2]

21 Feb 2025

Dear Qiaojie,

Thank you for your patience while we considered your revised manuscript "The zona incerta negatively regulates the red nucleus during movement cued by sound signals" for publication as a Research Article at PLOS Biology. This revised version of your manuscript has been evaluated by the PLOS Biology editors and the Academic Editor.

We are likely to accept this manuscript for publication, provided you satisfactorily address the remaining concerns as discussed via email, in particular the appropriate description of the software you used in your study. Currently, the manuscript mentions custom code, but you mentioned in your email that the code you used for data acquisition and analyses are open source code that was not generated by you. Please update the Code Availability Statement accordingly and mentions these software packages there and/or in the Methods section. Please also include references if possible.

We expect to receive your revised manuscript within one week. 

*Published Peer Review History*

*Press*

Sincerely,

Christian

Christian Schnell, PhD

Senior Editor

cschnell@plos.org

PLOS Biology

---

## [Editor Report · Decision Letter 3]

28 Feb 2025

Dear Qiaojie,

Thank you for the submission of your revised Research Article "The zona incerta negatively regulates the red nucleus during movement cued by sound signals" for publication in PLOS Biology. On behalf of my colleagues and the Academic Editor, Jennifer Bizley, I am pleased to say that we can in principle accept your manuscript for publication, provided you address any remaining formatting and reporting issues. These will be detailed in an email you should receive within 2-3 business days from our colleagues in the journal operations team; no action is required from you until then. Please note that we will not be able to formally accept your manuscript and schedule it for publication until you have completed any requested changes.

PRESS

Sincerely, 

Christian

Christian Schnell, PhD

Senior Editor

PLOS Biology

cschnell@plos.org